# Mechanical Properties of High-Performance Steel-Fibre-Reinforced Concrete and Its Application in Underground Mine Engineering

**DOI:** 10.3390/ma12152470

**Published:** 2019-08-03

**Authors:** Xiang Li, Weipei Xue, Cao Fu, Zhishu Yao, Xiaohu Liu

**Affiliations:** 1School of Civil Engineering and Architecture, Anhui University of Science and Technology, Huainan 232001, China; 2Post-doctoral Research Station of Safety Science and Engineering, Anhui University of Science and Technology, Huainan 232001, China

**Keywords:** HPSFRC, steel fiber, mechanical property, underground mine engineering, application research

## Abstract

In order to economically and reasonably solve the problem of mineshaft support in complex geological conditions, the mechanical properties of high-performance steel-fiber-reinforced concrete (HPSFRC) and its application in mineshaft lining structures were investigated in this study. Firstly, the mix proportion of HPSFRC for the mineshaft lining structure was obtained through raw material selection and preparation testing. Then, a series of mechanical property tests were conducted. The test results showed that the compressive, flexural, and tensile strengths of HPSFRC were 9%, 71%, and 53% higher than that of ordinary concrete, respectively. The fracture toughness of HPSFRC was 75% higher than that of the ordinary concrete and the fracture energy of HPSFRC was 16 times that of the ordinary concrete. Finally, the model test results of the HPSFRC shaft lining structure showed that the crack resistance, toughness, and bearing capacity of the shaft lining structure had been significantly improved under a non-uniform confining load because of the replacement of ordinary concrete with HPSFRC. HPSFRC was proved to be an ideal material for mineshaft support structures under complex geological conditions.

## 1. Introduction

It is necessary to build vertical shafts in stratum for ventilation, transporting minerals, personnel, materials, and equipment in order to exploit underground mineral resources by the well method. In order to maintain wellbore stability and resist formation pressure, it is necessary to build a strong support structure around the wellbore. In the bedrock section, ordinary concrete or reinforced concrete shaft lining structures have been usually used. However, with the increase of mining depth of coal mines, the engineering geological conditions of the strata which are passed through by the wellbore become more complex [1,2]. Especially, when the shaft construction passes through unstable strata, such as areas of well-developed small faults and stress concentration, weak fractured zones, and thick coal seams with large dip angles, shaft lining rupture accidents can occur even if a reinforced concrete shaft lining structure was adopted. Several coal mine vertical shaft structures have been damaged in weak fractured zones with complex stratum stress, such as the ventilation shaft of the Qujiang Coal Mine, the old auxiliary shaft in the Banji Coal Mine, and the ventilation shaft in the Guobei Coal Mine [3,4]. This problem seriously threatens safe production in mines.

Studies of stress analysis and the failure mechanism of shaft lining structures show that the stress state is very complex in unstable strata such as a fault fracture zone, where the stress state changes with time and stress concentrations are serious. The shaft lining structure is obviously subjected to non-uniform ground pressure. The concrete frequently used in such structures is easily damaged because it not only bears compressive stress but also tension and shear stress. This hidden danger always threatens the operation of the shaft. Therefore, finding or developing new support materials should be an effective technical way to avoid damage to mineshaft lining structures under complex geological conditions.

High-performance steel-fiber-reinforced concrete (HPSFRC) is a kind of composite material formed by adding a certain amount of steel fibers into a concrete matrix. Previous literature shows that tensile and shear strength of HPSFRC is 20%–70% and 20%–100% higher than that of ordinary concrete respectively [5,6,7]. It is especially suitable for mineshaft support engineering under complex geological conditions because of its strong toughness as opposed to ordinary concrete. Although HPSFRC has been widely used in civil engineering, it is mainly used in ground construction engineering and seldom in mineshaft support engineering under complex geological conditions. In order to economically and reasonably solve the problem of mineshaft support under complex conditions, it is necessary to thoroughly and systematically study the mechanical properties and characteristics of an HPSFRC shaft lining structure. Thus, mechanical property and model tests were conducted for this study.

## 2. Experimental Methods

The main goal of this study was to verify the superiority of HPSFRC in shaft lining structure. Thus, it was necessary to obtain the mechanical properties and the fracture toughness of HPSFRC and ordinary concrete by mechanical properties tests of the materials. Then, HPSFRC and ordinary concrete model tests need to be carried out to prove the superiority of HPSFRC shaft lining.

### 2.1. Mix Proportion

#### 2.1.1. Raw Materials

According to our extensive investigation, concrete with a standard cubic compressive strength of 60 MPa (C60) is usually used in the shaft lining structure of coal mines. The raw materials for the concrete specimens were obtained locally.

Conch P.II 52.5 Portland cement with a specific surface area of 389 m^2^/kg was selected. Its initial setting time was 145 min and final setting time was 199 min. Its stability was qualified. Its 3-day compressive strength was 25.7 MPa, and its flexural strength was 4.6 MPa. Its 28-day compressive strength was 55.8 MPa and flexural strength was 7.3 MPa.

Huaihe River sand was selected as the fine aggregate, with a fineness modulus of 2.9 and a saturated surface drying density of 2580 kg/m^3^. Basalt gravel was selected as the coarse aggregate, with a particle size less than 25 mm and a saturated surface drying density of 2720 kg/m^3^. The gradation of sand and gravel are shown in Figure 1 [8].

The high-performance powder water reducing agent NF, produced by Anhui Huaihe Chemical Co., Ltd. in Huainan, China, was selected. The water reducing rate was greater than 30%.

The slag produced by Qingya Building Material Co., Ltd. in Hefei, China, was selected, which had a specific surface area greater than 350 m^2^/kg. The microsilicon powder produced by Shanxi Dongyi Ferroalloy Factory (Xiaoyi, China) was selected, which had a specific surface area greater than 18,000 m^2^/kg.

RC65/60BN Dramix steel fiber, produced by Bekaert Applied Materials Technology Co., Ltd. (Shanghai, China), was selected. The appearance of the steel fiber is shown in Figure 2 and the parameters of the steel fiber are shown in Table 1 [9]. 

#### 2.1.2. Mix Proportion Test

The mix proportion of benchmark concrete was obtained through an orthogonal test (shown in Table 2). Ordinary concrete specimens named A-0 were arranged according to the mix proportion of benchmark concrete. HPSFRC specimens were made of steel fibers and benchmark concrete and were divided into three groups named A-0.5, A-1.0, and A-1.5 according to the volume fraction of steel fibers.

### 2.2. Mechanical Properties’ Test

#### 2.2.1. Specimens Preparation

The mixing sequence of HPSFRC was as follows. Firstly, cement, aggregate, and admixture were dried in a concrete mixer for 2 min. Secondly, steel fibers were added and stirred for 2 min. Then, NF powder water reducer was dissolved in 60% water and poured into the mixer for 3 min. Finally, the remaining 40% of water was poured into the mixer and stirred for 3 min.

We prepared four groups of specimens (A-0, A-0.5, A-1.0, and A-1.5) for each test for comparison. Each group contained three specimens. The dimensions and shapes of the specimens for the different tests are shown in Figure 3. 

The specimens were poured and vibrated with an 80-type concrete shaking table until the slurry was discharged. Immediately after shaping, the surface was covered with impervious film. Then, the specimens were placed in a standard maintenance environment (temperature: 20 ± 2 °C, humidity: ≥95%) for 24 h, then released from molds and put into a YH-40 standard maintenance box for 28 days.

#### 2.2.2. Test Method and Equipment

A 2000-kN universal testing machine (SFMIT, Changzhou, China) was used for compressive strength test, split tensile strength test, and wedge splitting strength test. A 1000-kN universal testing machine was used in flexural strength, and bending test. Load rate and displacement of platen were controlled and monitored by software. After all the specimens were removed from the standard maintenance box, they were wiped clean with a dry cloth, placed in the equipment, and corrected according to the requirements of the specifications. In each test, an anti-cracking net cover was installed around the testing machine to ensure safety [10,11].

The loading rate of the compressive strength test was controlled between 0.8 and 1.0 MPa/s. When the specimen was close to failure and began to deform sharply, the throttle of the testing machine was stopped from adjusting until failure occurred.

The loading rate of the flexural strength test was controlled between 0.08 and 0.10 MPa/s. The flexural test device was used to ensure that two equal loads could be simultaneously applied at three points of the specimen span.

The loading rate of the split tensile strength test was controlled at 0.08 and 0.10 MPa/s. A circular arc pad and a pad strip were placed between the upper and lower pressing plate and the specimen.

A small notch (20 mm deep) was cut at the bottom of the specimens of the bending test. The loading rate was controlled at 0.025 mm/min (rate of cylinder displacement). When the specimen was approaching failure, the loading speed was reduced appropriately.

In the wedge splitting test, the device was first clamped into the precut wide slot of the specimen (shown in Figure 4). A clamp extensometer was installed on the surface of the specimen. The graded continuous loading method was selected and the loading rate was controlled at 0.005 mm/s (rate of cylinder displacement) [12]. 

### 2.3. Model Test of Shaft Lining Structure

#### 2.3.1. Model Test Design

The Pansan Coal Mine belongs to the Anhui Huainan Mining Group. Its second auxiliary shaft lining structure is 8.0 m in internal diameter and 1004.2 m in depth. The structure passes through a fault fracture zone and two thick coal seams with gas outburst between 884 and 940 m depth. The rock mass in the fault zone is loose and fractured, and thick coal seams have to be drilled holes to extract gas. All of these would deteriorate the stress conditions of the shaft lining structure. As far as we know, several shaft lining structures have been damaged under similar engineering geological conditions. For the reasons above, the second auxiliary shaft lining structure of the Pansan Coal Mine was selected as the engineering background of the model test.

Due to the high strength and large size of the prototype structure, it was difficult to carry out the destructive test of the prototype shaft lining. The model must be smaller than that of the prototype and meet the conditions of a similar simulation.

It was necessary to find out the stress distribution on the cross-section of the model during the loading process and measure the failure load of the model. Therefore, the design of the shaft lining structure model should satisfy similar conditions of stress, deformation, and strength. According to the similarity theory and the basic equation of elasticity mechanics, the similarity index of the static shaft lining structure model was deduced by the equation analysis method [13].

According to the geometric equation, *C_ε_C_l_/C_δ_* = 1. According to the boundary equation, *C_p_/C_σ_* = 1. According to the physical equation *C_E_C_ε_/C_σ_* = 1, *C_v_* = 1. *C_ε_* is the strain similarity constant, *C_l_* is the geometric similarity constant, *C_δ_* is the displacement similarity constant, *C_p_* is the load similarity constant, *C_E_* is the elastic modulus constant, and *C_v_* is the Poisson’s ratio constant.

The HPSFRC shaft lining structure was a composite structure composed of several kinds of materials. The stress and deformation of the model had to maintain exact similarity to that of the prototype; thus, the geometry size of the model had to be similar to that of the prototype in the whole loading process. Therefore, we obtained *C_l_/C_δ_* = 1 and *C_ε_* = 1. Similar conditions of stress and deformation could be expressed as follows: *C_p_/C_σ_* = 1, *C_E_C_ε_/C_σ_* = 1, and *C_v_* = 1.

In order to ensure that the failure load and form of the shaft lining structure model were completely similar to that of the prototype, both the stress-strain relationship under the elastic state and the strength of the model should satisfy similar conditions. Firstly, the stress–strain curve of the shaft lining model and the prototype material should be similar in the whole loading process. Secondly, the strength of all parts should be similar. Finally, the strength criterion should be similar. Obviously, to satisfy all of the above similarity conditions, the material of the model must be the same as the original shaft lining. Thus, all of the other similarity constants were equal to 1, including the strength similarity constant (*C_R_*), reinforcement ratio similarity constant (*C_μ_*), and steel fiber content similarity constant (*C_ρ_*). Only the geometric similarity constant could be chosen according to the need.

According to our experimental conditions, we chose 3 as the geometric similarity constant, and the parameters of the shaft lining structure model are listed in Table 3. Two A-0 ordinary concrete and two A-0.5 HPSFRC shaft lining structure models of steel bars were designed for testing.

#### 2.3.2. Preparation of Model Specimens

Two inner ring steel bars and two outer ring steel bars with diameters of 10 mm were arranged in each model specimen. In the inner and outer rings, 178 vertical steel bars with diameters of 10 mm were evenly arranged.

Firstly, the specially made model templates were set up. Then, the steel bars which had been pasted with the test elements were arranged (shown in Figure 5). Thirdly, concrete was poured into the templates. After curing, the upper-end surfaces of the specimens were polished to ensure that the end models were smooth (shown in Figure 6).

#### 2.3.3. Equipment and Test Methods

The model loading test was carried out in a large-scale shaft lining structure test bed with 18 high-pressure cylinders of 1000 kN to simulate the horizontal load on the structure. Vertically, specimens were restrained by anchor bolts and tie rods, so that they were in a state of plane stress.

According to the Code for Design of Shaft and Chamber in Coal Mine [14], a large load is generally 1.1–1.2 times higher than a small load. Due to the eighteen high-pressure cylinders of 1000 kN in the test bed, the loading manner was designed as follows. Ten cylinders were loaded with large loads from east and west and eight cylinders were loaded with small loads from south and north (shown in Figure 7). The ratio of the large load to the small load was 1.2. Through the force analysis, the loading method was more disadvantageous than the sinusoidal non-uniform load used in the current design method of shaft lining. Therefore, the bearing capacity results of the test were safer than those of the standard design.

To monitor and analyze the stress and deformation of the lining model during the whole loading process, strain gauges and displacement observation elements were arranged inside and on the surface of the model. The layout of the element is shown in Figure 8.

## 3. Results and Discussion

### 3.1. Compressive Strength Test

Compressive strengths were calculated from the ultimate load values, and the numerical values were accurate to 0.1 MPa. The test results are shown in Table 4.

Table 4 shows that although the compressive strength of HPSFRC was higher than that of ordinary concrete, the maximum difference was no more than 10%. On the other hand, the compressive strength of HPSFRC with three kinds of steel fiber had little difference. According to the above two points, we learned that steel fibers contribute a little to the compressive strength of concrete. The conclusions are similar to other previous literature [15].

### 3.2. Flexural Strength Test

The results of flexural strength needed to be converted according to Equation (1), as the specimen size was nonstandard [10]. The numerical value was accurate to 0.1 MPa. The test results are shown in Table 5.
(1)$$f_{f}=0.85\frac{F_{V\max}l}{th^{2}}$$
where 0.85 is the dimension conversion coefficient (the conversion factor was obtained by comparing flexural strength of standard sample size with that of the non-standard size in this test).

Table 5 shows that the flexural strength of HPSFRC is much better than that of ordinary concrete. The flexural strength of HPSFRC increases with the increase of the steel fiber content. However, the extent decreases with the fiber content. The results are similar to the other previous literature [15].

### 3.3. Split Tensile Strength Test

The split tensile strength test results are shown in Table 6 and the test process is shown in Figure 9. 

Table 6 shows that the tensile strength of HPSFRC is significantly higher than that of ordinary concrete. The tensile strength of HPSFRC increases with the steel fiber content. Figure 9 shows that ordinary concrete fractured into two pieces by the load line during the test, while HPSFRC remained intact even after reaching the peak load. The conclusions are similar to the other previous literature [15].

### 3.4. Bending Test

According to the load, deflection, and crack opening displacement measured, the load–deflection curve (*F*–*δ* curve) was drawn, and the fracture toughness and fracture energy were calculated according to the following formulas [16]:(2)$$K_{IC}=\frac{6YM_{\max}\sqrt{a_{c}}}{th^{2}}$$
(3)$$G_{f}=\frac{A_{0}+mg\delta_{\max}}{t_{r}h_{r}}$$
where *Y* is a shape parameter. $Y=1.93-3.07(a_{0}/h)+14.53{\left(a_{0}/h\right)}^{2}-25.11{\left(a_{0}/h\right)}^{3}+25.8{\left(a_{0}/h\right)}^{4}$. Table 7 shows that the ultimate strength loads, peak deflections, fracture toughness, and fracture energy of HPSFRC were all higher than that of ordinary concrete and all of them increased with the steel fiber content. When the content reached 1.5%, the fracture toughness of HPSFRC was 75.0% higher than that of the ordinary concrete. And the fracture energy of HPSFRC was 16 times that of the ordinary concrete. The conclusions are similar to the other previous lectures [17,18,19].

Figure 10 shows that ordinary concrete had two stages in the bending test. One was the elastic stage and the other was the failure stage. The form of ordinary concrete revealed brittle failure characteristics. However, after the load reached the peak value, the form of HPSFRC revealed significantly more ductile behavior.

### 3.5. Wedge Splitting Test

Considering the complexity of the mineshaft lining structure, to fully reveal the difference of tensile strength and fracture toughness between ordinary concrete and HPSFRC, a wedge splitting test was conducted. After that, the *F*–*CMOD* curve was drawn. Then, tensile strength, instability fracture toughness, and fracture energy were calculated according to the following formulas [20,21]:(4)$$F_{H\max}=\frac{F_{V\max}}{2tg15{}^{\circ}}$$
(5)$$f_{t}=\frac{6F_{H\max}y}{t_{r}h_{r}^{2}}+\frac{F_{H\max}}{t_{r}h_{r}}$$
(6)$$E=\frac{1}{tc_{i}}[13.18{\left(1-\frac{a_{0}+h_{0}}{h+h_{0}}\right)}^{-2}-9.16]$$
(7)$$a_{c}=(h_{r}+h_{0})\left[1-\sqrt{\frac{13.18}{tECMOD_{c}/F_{H\max}+9.16}}\right]-h_{0}$$
(8)$$\alpha_{c}=\frac{a_{c}}{a_{0}+h_{r}}$$
(9)$$f(\alpha_{c})=3.675[1-0.12(\alpha_{c}-0.45)]{\left(1-\alpha_{c}\right)}^{-3/2}$$
(10)$$K_{IC}=\frac{F_{H\max}}{t_{r}\sqrt{h_{r}}}f(\alpha )$$
(11)$$G_{f}=\frac{\left(A_{0}+m_{0}gCOMD_{c}\right)}{2tg15{}^{\circ}t_{r}h_{r}}.$$

Table 8 shows that the tensile strength, fracture energy, fracture toughness, and instability fracture toughness of HPSFRC were all higher than that of ordinary concrete and all of them increased with the steel fiber content. Figure 11 shows the brittle failure characteristics of ordinary concrete and the ductility characteristics of HPSFRC. The conclusions are similar to the other previous lectures [18].

From the fracture morphology of the specimens in Figure 12, it can also be observed that the ordinary concrete was completely split into two parts. However, the steel fibers in HPSFRC transfer the tension stress after the specimens cracked.

The data of tensile strength, toughness, and fracture energy of the wedge splitting test had no comparability to that of the direct tension and bending tests. However, they also verified that HPSFRC had stronger tensile, toughness, and crack resistance than ordinary concrete. Further, the ductility characteristics of HPSFRC were similar to those of the bending test.

### 3.6. Ultimate Bearing Capacity of Shaft Lining Structure

Strength tests were carried out on four model specimens. The ultimate bearing capacity of the model specimens is listed in Table 9.

Table 9 shows that the bearing capacities of the HPSFRC models were higher than those of the ordinary concrete models, either in large or small load directions. The average bearing capacity of the HPSFRC model was 12.2%–12.7% higher than that of the ordinary concrete model under non-uniform load.

#### 3.6.1. Deformation of Model

The inner and outer edge strains of six sections were measured by strain gauges along the concrete surface. The strain variations in the middle section of the model specimen in the direction of large and small loads are shown in Figure 13.

Figure 13 shows that the tangential strain of the inner edge of the model was smaller than that of the outer edge in the large load direction. However, in the small load direction, the tangential strain of the inner edge was larger than that of the outer edge. Especially, the strain of the inner edge in the large load direction showed tension characteristics at the early stage of loading, which easily caused cracking in the ordinary concrete model. Therefore, HPSFRC is very advantageous to prevent rupture and improve the bearing capacity of a shaft lining structure under complex stress conditions.

#### 3.6.2. Failure Form of the Model

Due to poor ductility, the ordinary concrete model showed no obvious warning in the loading process, but in the end, it suddenly destructed. When destruction occurred, there was a loud noise, accompanied by massive concrete shedding. Local concrete was crushed, as shown in Figure 14. In the case of the HPSFRC model, the ductility was significantly improved (due to the addition of steel fibers), so that inclined microcracks appeared firstly when the model was damaged and then gradually expanded. When the load exceeded the shaft lining’s ultimate bearing capacity, the fracture surface was still connected by steel fibers (shown in Figure 15). Therefore, in shaft support engineering under complex conditions, the use of an HPSFRC shaft lining can greatly improve the deformation characteristics of the shaft lining and improve the safety of the project.

Under the action of a non-uniform load, the failure location of the shaft lining structure model was mainly located at the inner edge of the large load, where the concrete was subjected to eccentric pressure and bending moment, resulting in tensile stress, and local tensile cracks occurred during the loading process. Although the steel fibers hindered the rapid propagation of cracks, at the later stage of loading, the fracture location was still near that location. When the load was uniformly distributed, there was no bending moment and shear stress in all parts of the shaft lining, and the damage location was randomly distributed with the difference of construction quality.

#### 3.6.3. Stress Distribution of the Model

In the test, the model was always in the state of plane stress. When the model was in the stage of elastic deformation, the stress distribution of the inner and outer edges of the model conformed to the elastic theory. However, when the deformation transitioned from the elastic stage to elastoplastic or plastic stage, the constitutive relationship of concrete did not conform to Hooke’s law. For this reason, a single curve hypothesis was used for the steel bar. The deformation modulus and Poisson’s ratio were calculated by computer iteration and the stress values of the steel bar were obtained. Damage constitutive model was used for concrete according to “Code for Design of Concrete Structures” [22] and was also used for the HPSFRC according to Reference [23]. Thus, the stress curve of the main part of the model under the load could be obtained, as shown in Figure 16.

Figure 16 shows that the stress of the inner surface was always smaller than that of the outer surface in the cross-section of the model in the large load direction. On the contrary, the inner surface was always higher than that of the outside surface in the cross-section of the model in the small load direction. It is worth noting that tensile stress appeared in the inner surface of the model in the large load direction in the early stage, which was extremely unfavorable for ordinary concrete. According to the principles of elastic-plastic mechanics and rock mechanics, under the action of non-uniform pressure, the compressive stress of the concrete at the outer edge is concentrated while that of the concrete at the inner edge is just tensile, while in the direction of small load, it is just the opposite [24,25]. The tensile stress inevitably led to premature cracking of concrete and reduced the bearing capacity of the model. However, HPSFRC successfully improved the tensile and shear strength of the material and avoided premature cracking of the model in the early stage. Thus, the bearing capacity of the model was significantly improved.

## 4. Engineering Application

The above mentioned mechanical properties and model tests of the shaft lining structure showed that HPSFRC can evidently improve the bearing capacity of the shaft lining structure under non-uniform loads because of its remarkable tensile strength, and crack resistance. Therefore, this new type of shaft lining structure was used for an 884–940 m section of the second auxiliary shaft underground bedrock section of the Pansan Coal Mine, which was under complex stress conditions.

Short–excavation–short masonry technology with a 3.5 m cycle was adopted in the construction progress in the rock. When the shaft was excavated to an 884 m depth and entered the fault fracture zone, HPSFRC with steel bars was used to replace the original design of ordinary concrete with steel bars to construct the shaft lining structure of 800 mm thickness until the end of the 940 m depth.

In order to dynamically and continuously evaluate the safety of the new type of shaft lining structure, the stress distribution and time variation of the structure must be studied and analyzed. Therefore, concrete strain gauges were embedded in four perpendicular horizontal directions along the circumference of the shaft lining at a 940 m depth. The strain–time curve of HPSFRC was obtained by real-time observation, as shown in Figure 17.

Figure 17 shows that the strains of four directions in the HPSFRC shaft lining structure maintained stability. In addition, all of the strains were far less than the damage value of HPSFRC. The shaft lining structure ensures safe and reliable operation.

## 5. Conclusions

(1)The mechanical property tests showed that the compressive, flexural, and tensile strengths of HPSFRC were 9%, 71%, and 53% higher than those of ordinary concrete, respectively. Adding steel fiber can evidently improve the tensile and flexural strengths of ordinary concrete but has little effect on compressive strength.(2)With the increase of the steel fiber content, the fracture toughness and fracture energy of HPSFRC increased. When the steel fiber content reached 1.5%, the fracture toughness reached 0.42 which was 75% higher than that of the ordinary concrete, and the fracture energy reached 2003.9 N/m which was 16 times that of the ordinary concrete. The improvement effect was remarkable. Steel fibers in HPSFRC have shown excellent crack resistance in various tests, and HPSFRC shows remarkable ductility characteristics.(3)The model test showed that adding a certain amount of steel fiber to the concrete of the shaft lining structure had a significant effect on improving the bearing capacity of the shaft lining, even if resisting non-uniform loads.(4)In the shaft lining model test under a non-uniform load, the stress and strain of the outer surface in the direction of a large load and the inner surface in the direction of a small load were evidently larger than those in other directions or surfaces. Further, the tensile characteristics appeared in the inner surface of the large load at the initial stage of loading, which posed a hidden danger to the failure of the shaft lining model in the later stage.(5)Due to the poor ductility of ordinary concrete, the fracture of the shaft lining model of ordinary concrete was sudden and without any warning. However, the steel fiber in the HPSFRC shaft lining improved the brittleness and ductility of the concrete. When the shaft lining lost its ultimate bearing capacity, the fracture surface was still connected by steel fibers. Therefore, under complex engineering geological conditions, HPSFRC can greatly delay deformation, even if the lining cracks. The characteristics of HPSFRC can greatly improve the safety of shaft lining structures.(6)It has been proved by practice that the strain of HPSFRC tends to be stable, and the strain value is far less than the failure value of its material. The structure operates safely due to HPSFRC remaining intact and reliable. It avoids damage to the shaft lining, as with other deep wells under complex conditions with faults or high crustal stress. Thus, this new type of support structure has definite application value in deep wells.

## Figures and Tables

**Figure 1 materials-12-02470-f001:**
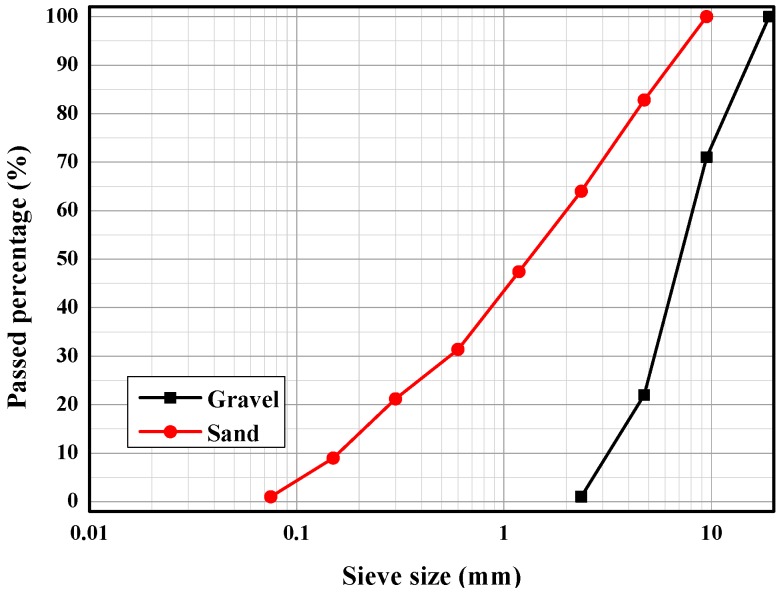
Aggregate gradation.

**Figure 2 materials-12-02470-f002:**
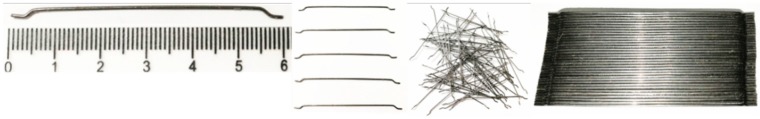
RC65/60BN Dramix steel fiber.

**Figure 3 materials-12-02470-f003:**
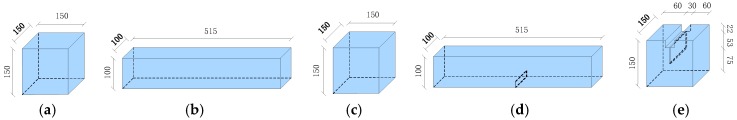
Dimensions and shapes of specimens: (**a**) compressive strength test; (**b**) flexural strength test; (**c**) split tensile strength test; (**d**) bending test; (**e**) wedge splitting test.

**Figure 4 materials-12-02470-f004:**
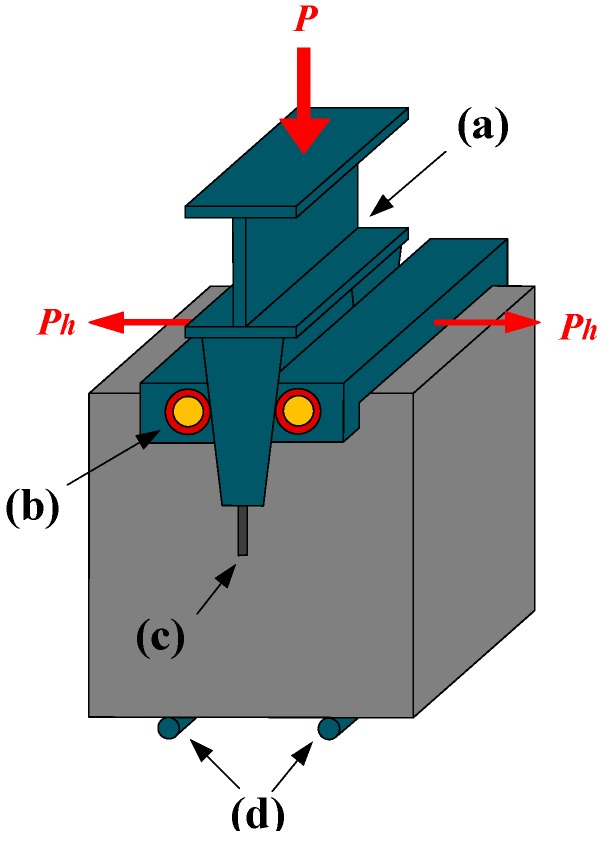
Loading manner of the wedge splitting test: (**a**) wedge loading fixture; (**b**) roller bearing; (**c**) notch; (**d**) hinge roller.

**Figure 5 materials-12-02470-f005:**
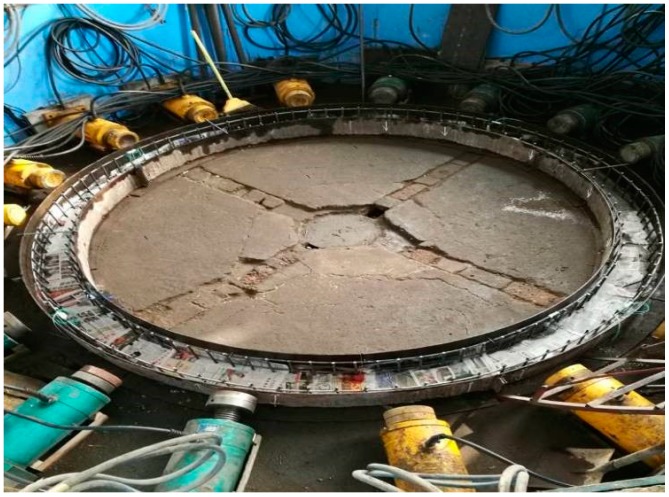
Templates for pouring shaft lining structure model.

**Figure 6 materials-12-02470-f006:**
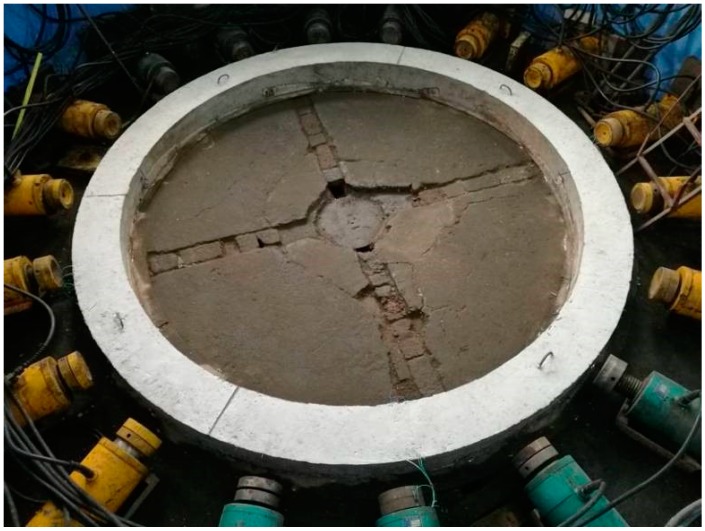
Shaft lining structure model with completed pouring.

**Figure 7 materials-12-02470-f007:**
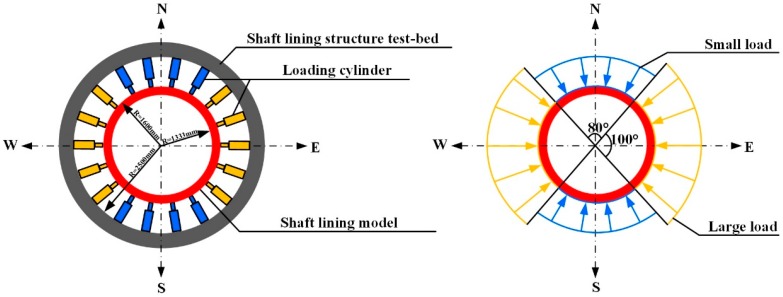
Load manner of shaft lining structure model.

**Figure 8 materials-12-02470-f008:**
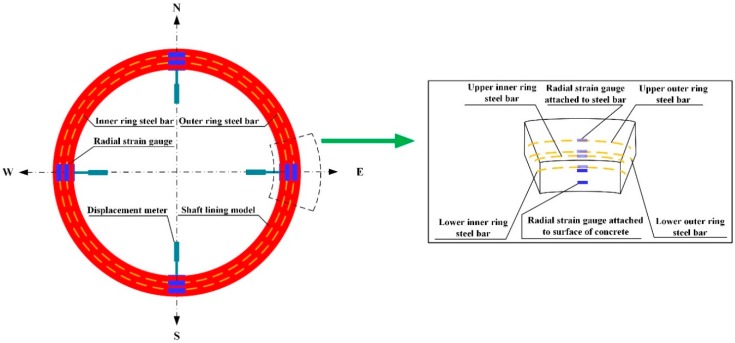
Placement of strain gauge and displacement meter.

**Figure 9 materials-12-02470-f009:**
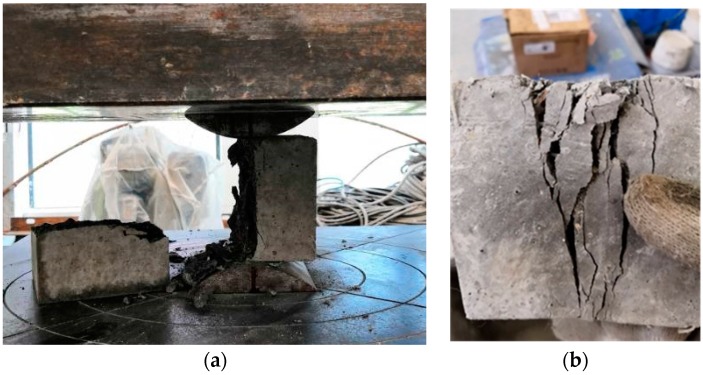
Specimens destroyed by split tensile strength test: (**a**) ordinary concrete specimen, (**b**) HPSFRC.

**Figure 10 materials-12-02470-f010:**
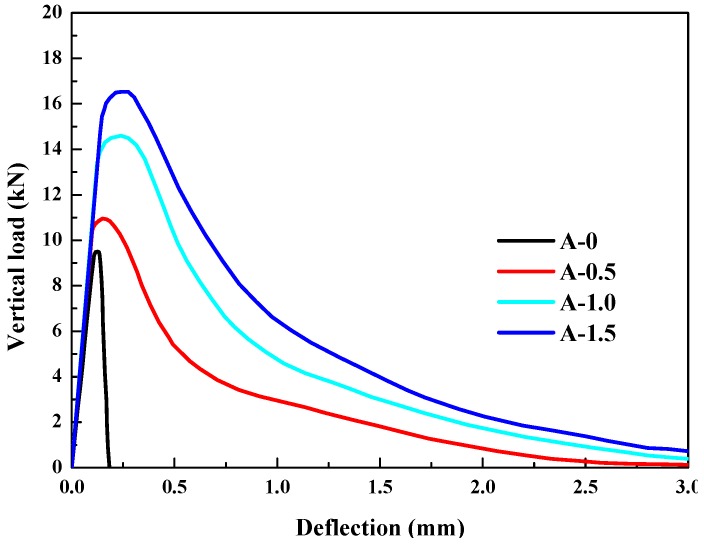
Relationship between vertical load and deflection.

**Figure 11 materials-12-02470-f011:**
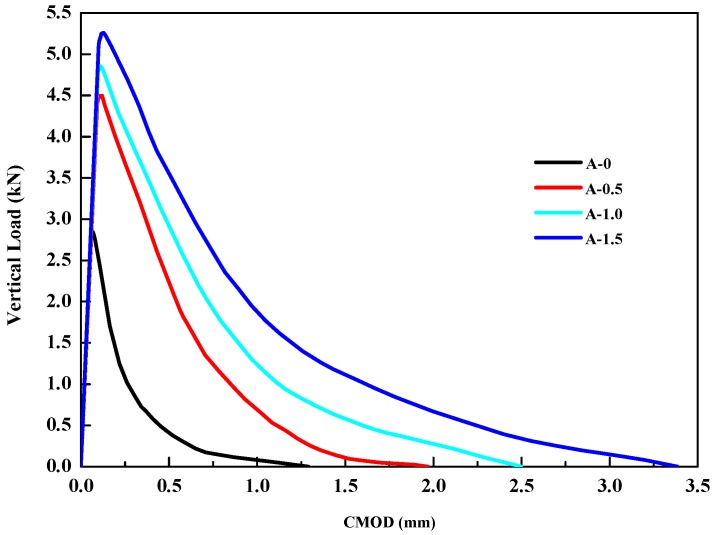
Relationship between vertical load and *CMOD* (Crack Mouth Opened Deflection).

**Figure 12 materials-12-02470-f012:**
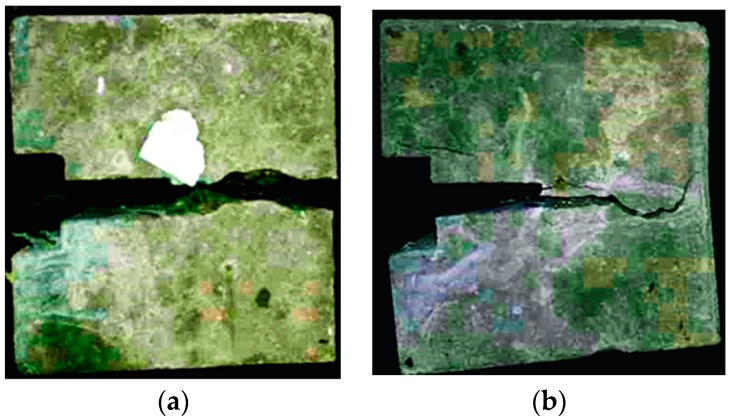
Specimens destroyed by the wedge splitting test: (**a**) ordinary concrete specimen, (**b**) HPSFRC.

**Figure 13 materials-12-02470-f013:**
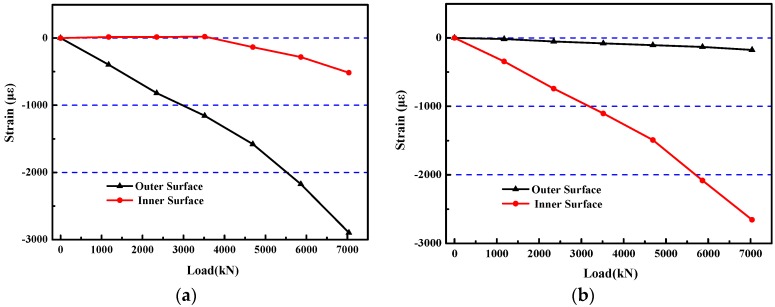
Relationship between strain and load of the S4 model: (**a**) large load direction; (**b**) small load direction.

**Figure 14 materials-12-02470-f014:**
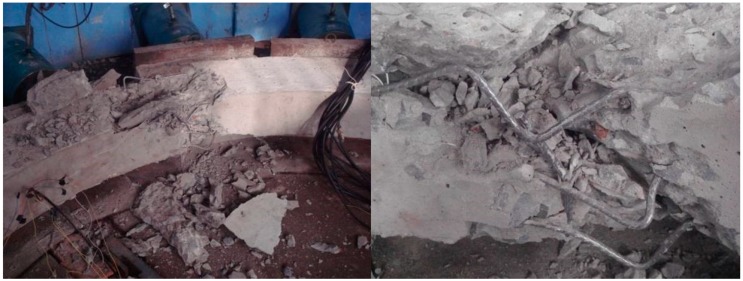
Failure form of model S1.

**Figure 15 materials-12-02470-f015:**
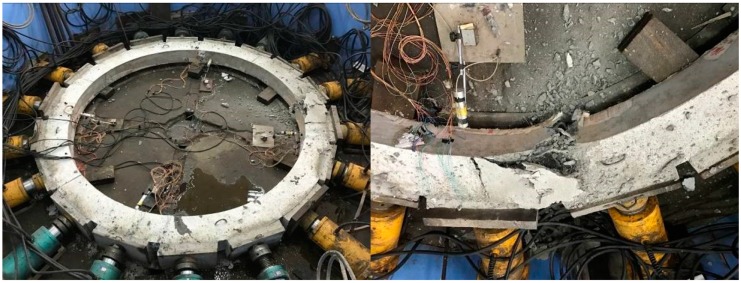
Failure form of model S3.

**Figure 16 materials-12-02470-f016:**
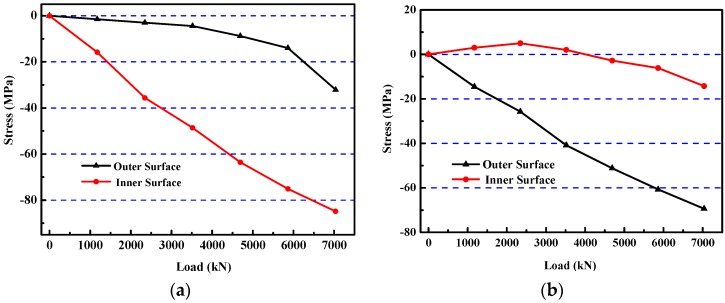
Relationship between stress and load of S4 model: (**a**) large load direction; (**b**) small load direction.

**Figure 17 materials-12-02470-f017:**
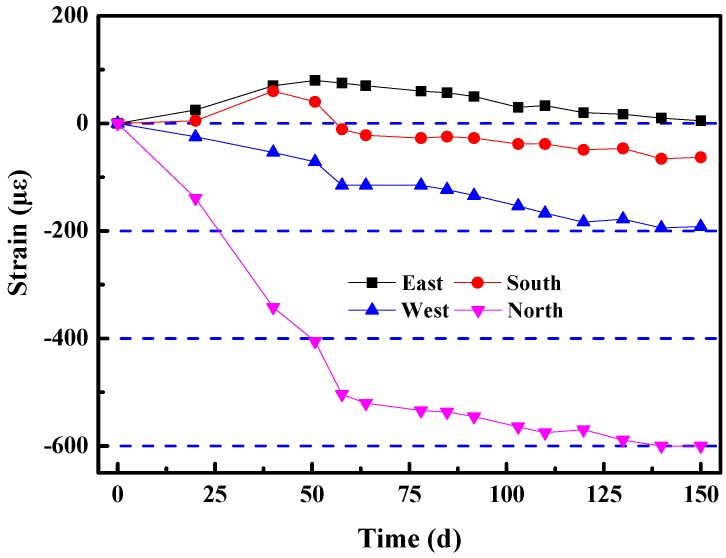
Relationship between strain and time of the HPSFRC shaft lining structure at a 940 m depth.

**Table 1 materials-12-02470-t001:** Characteristic parameters of steel fiber.

Length (mm)	Equivalent Diameter (mm)	Length–Diameter Ratio	Density (kg/m^3^)	Tensile Strength (MPa)
60	0.90	≥65	7800	≥1000

**Table 2 materials-12-02470-t002:** Mix proportion of benchmark concrete (kg/m^3^).

Cement	Sand	Gravel	Water	Plasticizer	Admixture	Slag
400	667.2	1088.5	164.3	7.2	105	25

**Table 3 materials-12-02470-t003:** Parameters of the shaft lining structure model.

External Diameter (mm)	Internal Diameter (mm)	Thickness (mm)	Height (mm)	Reinforcement Ratio (%)
3200	2666.6	266.7	210	0.56

**Table 4 materials-12-02470-t004:** Compressive strength test results.

Specimen Group	Compressive Strength (MPa)	Relative Increase (%)
A-0	64.3	0
A-0.5	69.6	8.2
A-1.0	69.3	7.8
A-1.5	70.0	8.8

**Table 5 materials-12-02470-t005:** Flexural strength test results.

Specimen Group	Flexural Strength (MPa)	Relative Increase (%)
A-0	6.1	0
A-0.5	7.7	26.2
A-1.0	9.2	50.8
A-1.5	10.4	70.5

**Table 6 materials-12-02470-t006:** Split tensile strength test results.

Specimen Group	Tensile Strength (MPa)	Relative Increase (%)
A-0	3.67	0
A-0.5	4.93	34.3
A-1.0	5.23	42.5
A-1.5	5.62	53.1

**Table 7 materials-12-02470-t007:** Bending test results.

Specimen Group	Ultimate Bearing Load (kN)	Relative Increase (%)	Peak Deflection (mm)	*K_IC_* $\left(\mathrm{MPa}/\sqrt{m}\right)$	*G_f_* (N/m)
A-0	9.6	0	0.1300	0.24	126.3
A-0.5	11.0	14.6	0.1562	0.28	1061.0
A-1.0	14.7	53.1	0.2182	0.37	1540.3
A-1.5	16.8	75.0	0.2231	0.42	2003.9

**Table 8 materials-12-02470-t008:** Wedge splitting test results.

Specimen Group	Wedge Splitting Tensile Strength (MPa)	Relative Increase (%)	$K_{IC}$ $\left(\mathrm{MPa}/\sqrt{m}\right)$	*G_f_* (N/m)
A-0	2.4	0	1.60	120.9
A-0.5	3.6	50.0	2.47	407.3
A-1.0	3.9	62.5	2.66	580.3
A-1.5	4.2	75.0	2.95	819.6

**Table 9 materials-12-02470-t009:** Bearing capacity of models tested under non-uniform load.

Model Specimen	Steel Fiber Content (kg/m^3^)	Bearing Capacity (kN)	
		Large Load	Small Load
S1	0	7271.7	6098.9
S2	0	7154.4	5864.3
S3	39	7975.4	6568.0
S4	39	8210.0	6919.9

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
