# Peer review of "Mechanical Properties of High-Performance Steel-Fibre-Reinforced Concrete and Its Application in Underground Mine Engineering"

_materials, 2019, doi:10.3390/ma12152470_

Round 1

Reviewer 1 Report

-   Figure 1. X axis must be constructed from left 0.01 mm to right 10 or 20 mm and with logarithmic scale.

-  Table 1. The value of  the density is 7800 kg/m3

Reviewer 2 Report

Thank you for improving the paper.

Author Response

Thank you very much for your valuable comments.

Reviewer 3 Report

The revised manuscript is significantly improved. 

Minor comments are made to the revised version of the manuscript (see attached pdf file). Please incorporate.

A general remark: When mentioning numerical calculations, it is necessary to provide more information about it. 

In a possible future study, it would be very useful to do some numerical modelling of the concrete shaft and try to better understand the obtained results as well as the real-case scenario. Furthermore, a wide range of different loading scenarios could easily be simulated.

After these comments have been incorporated, I would suggest to publish the paper.

Author Response

This manuscript is a resubmission of an earlier submission. The following is a list of the peer review reports and author responses from that submission.

Round 1

Reviewer 1 Report

This paper presents a study regarding mine shaft supports built with HPFC. The authors exposure. A serie of interesting tests to real scale with differents steel fibres content have been made. The deformations and the failure form of the model have been analyzed.The paper is well structured and the results provide valuable information

The paper could be accepted for publication in the journal after some changes that are listed below:

-  A figure with the size distribution of the aggregates used must be included.

-  Table 9, figure 13 and figure 16. There is amistake with the units. Load must be expressed in kN

-   Please use specimen instead sample and fibres instead fibers.

-   The results obtained must be compared with others similars papers of this field.

Reviewer 2 Report

The paper is interesting, using HPSFRC for in mine shaft lining structures is noteworthy, but achieve the quality of concreting with steel fibres in depth can be  difficult.

Reviewer 3 Report

The paper presents experimental results on HPSFRC. First, material characterization is performed to determine the optimum fiber content. Thereafter, a scaled structure was produced and tested under conditions expected in underground supporting structures.

Even though it is an interesting Topic, I would suggest to reject the paper for several reasons:

-Firstly, the paper lacks any literature review on both underground structures and steel fiber reinforced concrete. 2. The results obtained for material tests are partially fine (compressive strength, except for the possible high scatter - see Section 3.1). However, the results for all tensile tests do not correspond to other results found in the literature. It is irrelevant to show stress-time curves, the results should be presented as stress-deflection or stress-strain. SFRC should exhibit significantly higher ductility than found in the present tests. The valnues obtained for the fracture energy do not seem to correspond to the shown graphs. The difference in stiffness (bending test) for various fiber dosages indicated that there might be some major problem with the testing/measuring procedure, see figure 10.

-The measurement performed on the scaled model are not clear. For the same reason, the results are therefore also difficult to understand. In future, it is suggested to clearly indicate the measuring locations and direction, so that a better evaluation can be performed.

- Several conclusions are not supported by the results presented in the paper.
